# Heat-Stress-Mitigating Effects of a Protein-Hydrolysate-Based Biostimulant Are Linked to Changes in *Protease, DHN,* and *HSP* Gene Expression in Maize

**Irina I. Vaseva ***, **Lyudmila Simova-Stoilova, Anelia Kostadinova, Bistra Yuperlieva-Mateeva, Tania Karakicheva and Valya Vassileva**

Department of Molecular Biology and Genetics, Institute of Plant Physiology and Genetics, Bulgarian Academy of Sciences, Acad. Georgi Bonchev Str., Bldg. 21, 1113 Sofia, Bulgaria; lsimova@mail.bg (L.S.-S.); akostadinova_bg@yahoo.co.uk (A.K.); bistrayuperlieva@yahoo.com (B.Y.-M.); tania_karakicheva@abv.bg (T.K.); valyavassileva@bio21.bas.bg (V.V.)

\* Correspondence: vaseva@bio21.bas.bg or irina.vaseva@abv.bg

**Abstract:** The growth-promoting and heat-mitigating effects of a commercially available protein-hydrolysate-based biostimulant, Kaishi, during the early vegetative stage was investigated by applying it as a foliar spray on soil-grown maize plants or in the nutrient solution of hydroponically grown plants. At $10^{-3}$ dilution, the biostimulant inhibited germination and delayed the growth progress, while at $10^{-6}$–$10^{-12}$ dilutions, it promoted shoot and root growth. Heat stress caused biomass reduction, decreased leaf pigment content and the chlorophyll *a*/chlorophyll *b* (chl *a*/*b*) ratio, caused starch depletion, and increased lipid peroxidation. Kaishi priming resulted in the substantial mitigation of negative stress effects, maintaining growth, stabilizing pigment content and the chl *a*/*b* ratio, restoring the leaf starch content, lowering the malondialdehyde (MDA) level, and significantly increasing the free proline content. The expression profiles of a set of genes coding for heat shock proteins (HSPs), dehydrins (DHNs), and proteases were analysed using qRT-PCR after heat stress exposure. The biostimulant-treated plants had higher transcript levels of certain *HSPs*, *DHNs*, and protease-coding genes, which remained stable or increased after the applied stress. The results demonstrate that very low concentrations of the biostimulant exerted stress-mitigating effects that could be linked to organ-specific changes in the gene expression of certain stress-inducible proteins.

**Keywords:** dehydrins; heat stress; HSPs; proteases; protein-hydrolysate-based biostimulants; *Zea mays* L.





## 1. Introduction

Along with wheat and rice, corn is one of the major monocotyledonous staple crops with great economic importance. Since it requires relatively high temperatures at the early vegetative stage, it should be sown in late spring. The major maize-producing regions in the world occupy territories with a moderate or subtropical climate and average late spring day temperatures ranging from 16 °C (continental parts of the USA, Asia, and Europe) to 30 °C (countries from the Mediterranean region, Mexico, sub-Saharan Africa, and India) (https://worldweather.wmo.int/en/home.html, accessed in 15 January 2022). With climate change leading to overall warming, these average baseline values could increase substantially over the coming decades, and the occurrence of heat waves will probably become more frequent and more intense [1–4]. A recent example is the record-setting heat waves documented in June 2021 for a number of territories in the USA (https://yaleclimateconnections.org/, accessed in 15 January 2022).

It is known that the maize maturation stage is the most susceptible to heat stress, resulting in significant yield loss [2,5–7]. The negative consequences of elevated temperatures during the early vegetative stage have been relatively less studied. Air temperature is a

limiting factor in the geographical distribution of corn, and temperatures above 35 °C, especially with the combination of hot and dry weather, have a negative effect on pollination and fertilization [3]. The optimal ambient temperature for maize germination is 10–12 °C, while the growth and development of the vegetative organs usually require 18–20 °C [6]. Maize is known to be very sensitive to drought during the vegetative developmental stage [8]. Therefore, the effect of heat weaves, which combine high temperature and dehydration stress, could be very damaging for the cultivation of this crop. Recent reports on the more frequent occurrence of heat events [1,4] coinciding with the early vegetative development of maize plants call for a detailed evaluation of the response to heat stress during the early vegetation stage and the introduction of efficient plant breeding approaches to cope with the consequences.

Biostimulants are substances of natural origin that support the pro-ecological cultivation of vegetables and fruits [9]. Although some products of this class may contain some nutrient elements, their amounts are too small to have a fertilizing effect [10]. Plant biostimulants are complex formulations of different ingredients, as the combinations were found to trigger more valuable systemic effects than the ones obtained after the application of the individual components alone [9,11]. The ability of biostimulants to exert a positive impact on plant growth and productivity without any hazardous consequences has made an aggressive entrance on the market as an environmentally safer and cost-effective approach for improving crop fitness. However, the routine use of biostimulants in agriculture as stress-mitigating compounds remains very limited due to the insufficient information regarding their mode of action under different stress conditions, which could be crop-specific and also dependent on the application methods, timing, and concentration rates [9,12]. Generally, very low amounts of biostimulants are able to evoke natural plant defence, which further increases their practical value and relevance for the agricultural sector [13].

The biostimulant Kaishi® is assigned to the category of protein hydrolysates, according to the classification proposed by du Jardin (2015) [14]. It contains organic nitrogen and 12% free L-amino acids of vegetal origin obtained via enzymatic hydrolysis. The beneficial action of protein-hydrolysate-based biostimulants has been attributed to growth-regulating effects [15–17], the preservation of photosynthetic apparatus [17–19], the maintenance of redox homeostasis [20–22], stabilizing effects on enzymes [17], and the adjustment of gene expression, especially for genes involved in nitrate transport and the metabolism of reactive oxygen species [23]. It has been suggested that this class of biostimulants could interfere with hormonal activity, regulate ion transport [15], and directly stimulate carbon and nitrogen metabolism [15,24–26]. According to the manufacturer's instructions, Kaishi® should be administered as a foliar spray usually after the canopy has experienced abiotic stress. A recent study of the same product demonstrated that soybean seed pre-treatment is also efficient in being able to modify the amounts of some bioactive compounds, also affecting leaf structural and photosynthetic traits [27].

Protein-hydrolysate-based biostimulants are applied both as foliar sprays and through drip irrigation systems, as it is known that plants are able to absorb amino acids both through roots and through leaves [15]. It is well known that the application of amino acids positively influences nitrogen metabolism in crops and increases their productivity, especially when administered as a seed pre-treatment [28]. For example, the ability of free amino acid preparations to ameliorate the negative effects of high salinity [29] and freezing and cold stress in different crops [20,30] has been previously reported. Research on the heat-stress-mitigating properties of protein-hydrolysate-based biostimulant Terra-Sorb foliar® in ryegrass (*Lolium perenne* L.) subjected to high temperatures has shown that the priming resulted in improved photosynthetic efficiency and higher chlorophyll and carotenoids levels [18]. The application of a biostimulant product containing amino acids as a priming agent has been shown to increase tolerance to heat stress in cucumber plants [21].

Heat stress (HS) is harmful to plants, causing ion and nutrient imbalance, direct injuries to membranes and macromolecules, and indirect damage from secondary oxidative and osmotic stress [31,32]. According to transcriptomic data, protein renaturation, biomembrane

repair, osmotic adjustment, and redox balance play key roles in the response of maize plants to HS [32]. To counteract protein denaturation and membrane damage under HS, plants activate the synthesis of specific proteins, such as heat shock proteins (HSPs) and dehydrins (DHNs); in addition, various proteases are mobilized to remove irreversibly damaged proteins [33].

Plant HSPs are essential molecular components involved in thermotolerance, playing an important role in survival under extreme conditions. They have been classified into five major groups: HSP 100, HSP 90, HSP 70, HSP 60, and small HSPs (sHSPs), based on their molecular mass [34]. Plants have particularly high numbers of diverse small HSPs (15–40 kDa), which function as molecular chaperones involved in sustaining the normal folded state of proteins and preventing their inactivation in co-operation with HSP 70 and HSP 101 [35]. Transcriptomic studies report the differential expression of several *sHSP* genes upon the foliar application of protein hydrolysates on maize seedlings [16]. Six *HSP* genes were identified to be significantly upregulated during heat stress in the heat-tolerant maize lines, indicating that they play a role in stress protection [36].

Dehydrins (DHNs) belong to the large group of extremely hydrophilic proteins known as Late Embryogenesis Abundant (LEA) proteins [37]. They are glycine-rich and express low secondary structure in vitro but they often gain structure when bound to a target [38,39]. A distinct molecular feature of DHNs is the presence of a consensus sequence, rich in Lys-residues, known as the K-segment (EKKGIMDKIKELLPG). At the N terminus of the amino acid chain, a Y-segment (V/TDEYGNP) could be found as well. Some dehydrins also contain serine-rich tracts (the S-segment) that can be modified by phosphorylation and bind ions. The number (n) and order of the Y-, S-, and K-segments define five different DHN sub-classes: YnSKn (alkaline), SKn (acidic), Kn, YnKn, and KnS [37]. DHN accumulation is associated with a tolerance mechanism leading to the maintenance of cellular turgor, the protection of biomolecules, and the stabilization of cell structures and membranes. For some dehydrins with metal-binding capacity, even a reactive oxygen species (ROS) scavenging function under different stress conditions has been reported [39]. A strong association between DHNs and plant tolerance to heat stress has been found [31]. Biostimulant application results in significantly higher levels of the drought-responsive DHN-like proteins in tomato plants exposed to water stress [40].

Proteases are major components participating in the processes of protein breakdown and recycling provoked by adverse environment. They are also involved in the normal plant growth and development program [41]. These enzymes catalyse the hydrolysis of peptide bonds and show high structural and functional diversity. Promoter regions of the genes encoding cysteine-type proteases in maize are enriched in stress-responsive regulatory elements, suggesting the active involvement of these proteases in the processes that counteract abiotic stress [42].

This study aimed at elucidating whether the protective effect of the biostimulant Kaishi on maize plants subjected to heat stress could be linked to the activation of the defence mechanisms overcoming direct injury, such as the enhanced expression of HSPs, DHNs, and proteases. The presented case study is based on the application of the diluted biostimulant as a priming agent in maize with good mitigating properties against heat stress during early vegetation. We investigated the protective potential of the biostimulant Kaishi in two experimental models. In the first setup, dilution series of the protein-hydrolysate-based biostimulant were applied as foliar spray on young soil-grown maize plants. In the second experimental setup employing hydroponically grown plants different dilutions of the product were applied directly in the growth media. The possible molecular mechanisms behind the biostimulant's stress-protective effects were assessed on the hydroponically grown plants via organ specific transcript profiling of DHN-, HSP-, and protease-coding genes.

## 2. Materials and Methods

The experiments were performed with the maize hybrid Knezha 307 (obtained from the Maize Research Institute, Knezha, Bulgaria) in two experimental models: soil-grown

plants experiencing simulated heat weaves (exposure to 40 °C for 5 h during 5 consecutive days), and hydroponically grown plants subjected to heat shock (single high-temperature treatment at 45 °C for 1 h).

### 2.1. Soil Experiment

The protective potential of the biostimulant Kaishi® (Sumi Agro, Paris, France) against simulated heat waves was investigated with the dilution series $10^{-3}$, $10^{-6}$, $10^{-9}$, and $10^{-12}$ applied as a priming agent in the form of foliar spray on the young soil-grown plants. The information on the biostimulant composition provided by the manufacturer states that it contains 12% $w/w$ free L-amino acids (standard aminogram: L-glutamic acid, L-aspartic acid, L-alanine, L-arginine, L-cystine, L-phenylalanine, L-glycine, L-histidine, L-isoleucine, L-leucine, L-lysine, L-methionine, L-proline, L-serine, L-tyrosine, L-threonine, L-tryptophan, and L-valine), none of which exceeds 20% of the total amount. The product contains organic nitrogen 2.0% $w/w$ (with total nitrogen estimated to be 2.0% $w/w$). As a starting concentration, we used an amount (1 mL/L or $10^{-3}$ dilution which corresponds to 0.001% solution) that approximated the range recommended by the manufacturer (1–3 L/ha). After 72 h of germination in darkness, equally developed seedlings (5 individuals per pot of 400 g capacity) were transferred on leached meadow cinnamonic soil (pH 6.2). The plants were grown under controlled conditions (day/night temperatures of 25 °C/21 °C, 150 µmol m$^{-2}$ s$^{-1}$ photosynthetically active radiation, and 16 h/8 h light/dark photoperiod) until the first true leaf was completely expanded, which corresponded to 6 days after the transfer on soil (6 DAS). Soil moisture was controlled with gravimetric measurements of the pots. Water was added daily to maintain the relative soil humidity of 70% of the maximal soil moisture capacity. At 7 DAS, the plants were sprayed with different dilutions of the biostimulant ($10^{-3}$, $10^{-6}$, $10^{-9}$, or $10^{-12}$) or with distilled water (control group), with Tween 20 (0.5% $v/v$) used as a surfactant, until complete leaf coverage (approximately 10 mL of solution was used per pot). The simulation of heat weaves followed criteria described in [4]. The "heat wave" treatment lasted for 5 consecutive days starting at 8 DAS. Half of the plants from the different treatment groups were transferred for 5 h (starting at midday) in a "hot" growth chamber, where the temperature was maintained at 40 °C. The other parameters in the "hot" growth chamber (light intensity and air humidity) remained unchanged. After the high-temperature treatment, the pots with the stressed plants were transferred back to control growth conditions. The biostimulant application of the tested dilutions was repeated on the third day of the stress program. The heat-treated plants did not receive watering during the stress period, mimicking the incidence of plant exposure to combined heat and drought stress. The soil humidity of the controls was maintained during the whole experimental period, while it dropped down to 35–37% in the heat-stressed variants at the end of the treatment.

At 13 DAS, the watering of the heat-stressed plants was resumed. Malondialdehyde (MDA), free proline content (L-Pro), and photosynthetic pigments (carotenoids and chlorophyll $a$ and $b$) in the tissues of the plants from the different experimental groups were assessed 24 h later (at 14 DAS). The biometric measurements were taken after 72 h of recovery (16 DAS). The in-gel staining of antioxidant enzyme activities and metabolite content analyses were performed on material from the same sampling point (16 DAS). The different treatments were designated as follows: C, controls; H, heat-stressed; K$10^{-3}$, K$10^{-6}$, K$10^{-9}$, and K$10^{-12}$, plants grown under normal conditions but treated with the respective Kaishi amounts; HK$10^{-3}$, HK$10^{-6}$, HK$10^{-9}$, and HK$10^{-12}$, plants subjected to heat stress but treated with the respective amounts of the biostimulant Kaishi.

### 2.2. Hydroponic Experiments

2.2.1. Biostimulant Effects on Germination and Growth of Hydroponically Grown Maize Plants

Germination and continuous growth tests in the presence of different amounts of the biostimulant (K0, K$10^{-3}$, K$10^{-6}$, K$10^{-9}$, and K$10^{-12}$) were performed to elucidate the most

appropriate concentration for the experiments with hydroponically grown plants. During the first 72 h, the Petri dishes with germinating seeds on filter paper soaked with different Kaishi concentrations were kept in darkness at 22 °C. The primary growth (shoot and root elongation) was documented, and the plants were transferred to plastic containers filled with different biostimulant dilutions in deionized water. The primary root and shoot lengths were recorded in dynamics until the 7th day from the transfer on liquid media under normal growth conditions (16/8 h day/night photoperiod, 25 °C/21 °C day/night temperature, and 150 $\mu$mol m$^{-2}$ s$^{-1}$ light intensity). The measurements of the organ length were performed with ImageJ 1.52r.

2.2.2. Heat Shock Applied on Hydroponically Grown Plants Pre-Treated with Biostimulant

The biostimulant's stress-protective effect was tested in primed, hydroponically grown plants subjected to heat shock at 45 °C for one hour. The nutrient solution contained $10^{-12}$ dilution of the biostimulant Kaishi. Briefly, equally germinated seedlings were transferred in $1/2$ litre containers with distilled water (x 24 individuals per one vessel) which were kept under normal conditions (16 h/8 h light/dark photoperiod, at 25 °C/21 °C day/night temperature, and 150 $\mu$mol m$^{-2}$ s$^{-1}$ light intensity) for four days (the age of the plants corresponded to 7 days after germination, or 7 DAG). On the 5th day of the photoperiod growth (at 8 DAG), some of the plants were transferred on $10^{-12}$ Kaishi solution. Twenty-four hours later (at 9 DAG), some control plants as well as some of the biostimulant pre-treated individuals were subjected to heat shock (45° C) for 1 h in a "hot" growth chamber. The samples for RNA extraction were collected 24 h after the heat stress treatment (at 10 DAG). The organ length, fresh weight (FW), and dry weight (DW) of the individuals from the different treatment groups and the organ-specific protease activities were measured 48 h after the applied stress (at 11 DAG). The different treatments in this experiment are designated as follows: C, controls; HS, heat-stressed; K, biostimulant-treated but grown under normal conditions; K-HS, biostimulant pre-treated and subjected to heat stress.

*2.3. Determination of Photosynthetic Pigments, Oxidative Stress Markers, and Metabolites*

Spectrophotometric analyses of chlorophyll *a* (chl *a*), chlorophyll *b* (chl *b*), carotenoids (Car), malondialdehyde (MDA), L-Pro, phenolics, total flavonoid content, and soluble sugar and starch were performed following the procedures described in [43]. In brief, the plant material (100 mg flash-frozen in liquid nitrogen) derived from the second true leaf (soil-grown plants) was homogenized in 80% (*v/v*) ethanol, and the absorbance at 470 nm, 649 nm, and 664 nm was recorded to calculate the content of photosynthetic pigments. MDA was determined in the same sample as thiobarbituric acid-reagent product by measuring the absorbance at 440 nm, 532 nm, and 600 nm. The content of L-Pro in the sample was measured in a reaction mix containing 150 $\mu$L of extract and 100 $\mu$L of 1% (*w/v*) ninhydrin in 60% (*v/v*) acetic acid and 20% (*v/v*) ethanol. The absorption was read at 520 nm, and L-Pro content was calculated according to the prepared standard curve. Soluble sugar and starch content was estimated using Anthrone reagent, reading the absorbance at 625 nm. The total phenolic content was determined by employing Folin–Ciocalteu reagent at 720 nm. The total flavonoid content was measured using the AlCl$_3$ protocol. A Multiskan Spectrum UV/VIS spectrophotometer (Thermo Fisher Scientific, Vantaa, Finland) was used for the absorbance measurements.

*2.4. Enzyme Assays*

The plant material for the enzymatic assays (0.8 g root and 0.5 g leaf samples derived from the second and the third true leaves) was quick-frozen in liquid nitrogen and stored at −65 °C until it was analysed. The protein extraction was performed with 2.5 mL of 100 mM Tris-HCl buffer pH 7.5, containing 150 mM KCl, 10 mM CaCl$_2$, 5mM cysteine, and 50 mg of Polyclar AT. After centrifugation for 30 min at 4 °C and 14,800 rpm, sucrose was added to the supernatant (20% *w/v* final concentration), and the mixture was aliquoted for the subsequent analyses. Protein content in the samples was estimated using the method

of Bradford [44]. Superoxide dismutase (SOD) isoforms were separated via electrophoresis in 10% resolving and 4% stacking gel at 4 °C under non-denaturing conditions followed by the activity staining protocol according to [45]. Unspecific peroxidase (POX) activity staining was performed with benzidine according to [46] after separation in 7% resolving and 3.75% stacking gel at 4 °C under non-denaturing conditions. Catalase (CAT) activity was estimated after separation in 7% resolving and 3.75% stacking gel at 4 °C under non-denaturing conditions and the activity staining followed the procedure described in [47]. Proteolytic activities were resolved according to [48] using 10% separating gel co-polymerized with 0.1% gelatine and 5% stacking gel. Protein loading was performed in sample buffer (without boiling the samples). After electrophoretic separation at 4 °C, SDS was removed using 2% TX-100. The gels were incubated for 22 h at room temperature in 50 mM $NaCH_3COO$ (pH 5.0) supplemented with 5 mM cysteine, or in 50 mM Tris-HCl (pH 7.5) containing 150 mM KCl, 10 mM $CaCl_2$, and 5 mM cysteine, and subsequently stained with colloidal Coomassie. The in-gel enzyme activity staining assays were repeated twice. The analysis of the gel images was performed with Image J software to estimate the total enzyme activity.

### 2.5. RT-PCR Analysis

Total RNA was extracted using a GeneJET Plant RNA Purification Kit (Thermo Scientific, Waltham, MA, USA). The isolated RNA was quantified using a Nano Drop 2000 spectrophotometer (Thermo Scientific). The synthesis of cDNA was performed via the reverse transcription of 1 µg of RNA with the iScript cDNA synthesis kit (Bio-Rad, Hercules, CA, USA), according to the manufacturer's instructions. The real-time quantitative RT-PCR (qRT-PCR) analyses were performed with AccuPower® GreenStar™ qPCR PreMix (Bioneer, Daejeon, South Korea) in three technical replicates with a 'PikoReal' Real-Time PCR System (Thermo Scientific).

The following settings were used in the qRT-PCR runs: 95 °C for 15 min and 45 cycles of 95 °C for 10 s followed by 55 °C–60 °C for 30 s and final melting curve analysis with a temperature range of 60 °C–95 °C in 0.2 °C increments for 60 s. The relative expression of the target genes was calculated with the ΔΔCq method [49] using elongation factor 1-alpha (*αEF1*, LOC542581), actin 1 (*ACT1*, LOC100282267), and alpha tubulin 5 (*TUB5*, LOC542248) as reference genes. The primer sequences used in the expression analyses are presented in Table 1.

**Table 1.** Primer pairs used in the qRT-PCR analyses.

| Gene Name | Locus | Forward PRIMER (5′–3′) | Reverse primer (5′–3′) |
|---|---|---|---|
| *ZmDHN1* | LOC542373 | agggacagggacagtttcct | ccactcgcaagtgctgtacta |
| *ZmDHN2* | LOC542251 | cgatcagaagcgttgcgttg | ggtctttaaagcacacgggc |
| *ZmDHN13* | LOC100285266 | cacaaggaaggcatcgtgga | gctgcgacaccagatctcag |
| *ZmDHNXero1* | LOC100279027 | cgtgcatattgctgtgctcc | agccagagccaaacctacac |
| *ZmDHNCOR410* | LOC100281087 | gaaggtagctagcgttggca | accacgtcctacacaagcag |
| *ZmDHN4* | LOC103635599 | cgccacaggcatatctggaa | tccttcaggcccttcttcg |
| *ZmHSP1* | LOC100286044 | cggagaacaccaaggtggat | accttccacgtccaatcgtc |
| *ZmHSP16.9* | LOC100280576 | cccaacccaatcccaatcca | cacggagaatgggtcgaaca |
| *ZmHSP18a* | LOC542293 | agttcatgcgcaagttcgtg | gacaacggtctcccctcag |
| *ZmHSP22* | LOC100283239 | cgaagaagagtattggcggc | gatcgcacactttctctgcc |
| *ZmHSP26* | LOC542576 | ccaagtagcgaaatggcagc | gtcgacactgttgtccctgt |
| *ZmHSP70* | LOC103635762 | agccgatgatcgtggttagc | ttgaaataggcaggcacggt |
| *ZmSUMO* | LOC100280713 | tgcaggagaatggatgggac | gtcctcctgcggaagtagtg |
| *ZmSBT2.1* | LOC100381627 | tctttgggtgttctcgcctc | cggcaaattaatggcgaggg |
| *ZmSEN102* | LOC100280695 | gagaatggctacgtgcggat | acaccgtctcgttgagttgt |
| *Zmprot2* | LOC100281516 | cgtcatgtccgatgtcaagc | gatacgggacgcctacagtg |
| *ZmCysprot1* | LOC100283826 | gcatgaggacctcgatctgg | tacagcggattcatgggacg |
| *ZmSAG39* | LOC103641507 | tagtggactgcgacgtgaac | tcctcgtagcccttgatgga |
| *Zm ACT1* | LOC100282267 | cttcgaagaaaatgcggcgg | attctgctgaagaggtggc |
| *Zm TUB5* | LOC542248 | cctgcccaaggcaagagaaa | gaggaatcactgggcatggt |
| *Zm αEF* | LOC542581 | tgttctcactctcagacaccag | cccatggctgaaggaaaatgt |

*2.6. Statistical Analyses*

The results are based on a completely randomized experimental design using three biological replicates per experimental group. At least fifteen individuals per treatment were used for the measurements (FW, DW, and organ length). The growth parameters, biochemical data, and the measurements of the staining intensities in the in-gel enzyme assays were subjected to multifactor ANOVA (multiple range test) using the program StatGraphics Plus 2.1. Statistical analyses of the chl *a*/chl *b* ratio, measurements of the growth parameters of hydroponically grown maize plants subjected to heat shock, and the expression levels in the transcript profiling were performed in Excel (one-way ANOVA). The error bars in the graphs reflect the standard deviation (SD) or the Standard Error (SE).

## 3. Results

*3.1. Effects of the Biostimulant on Soil-Grown Maize Plants Subjected to Simulated Heat Wave Treatment*

### 3.1.1. Growth Parameters

The heat-stress-mitigating properties of the biostimulant were tested using a dilution series starting with the amount recommended by the producer, which approximates $10^{-3}$ dilution in our experiments. The manufacturer's instructions advise to use the product after the plants have experienced unfavourable environmental conditions. In our experimental setup, we applied the product as a foliar spray prior to the simulated heat weaves and in the middle of the stress period to test whether a priming strategy could be beneficial for better survival and more efficient recovery of the affected plants. A good heat-stress-mitigating effect was achieved with the application of the $10^{-9}$ dilution of Kaishi. The plants from this treatment group exhibited the least negatively affected organ growth (Figure 1A) and reductions in FW and DW (Figure 1B).

The plants treated with the $10^{-12}$ dilution accumulated higher shoot biomass under normal conditions, and this was also observed in the pre-treated heat-stressed individuals which received the same biostimulant application (Figure 1B). The effect was less pronounced in the shoot dry weight measurements. Kaishi pre-treatment had a weaker effect on root growth under heat stress. Some positive trends were observed in the roots of the individuals that received the $10^{-9}$ dilution of the biostimulant before the high temperature treatment, while the $10^{-6}$ concentration had a slight root growth-inhibiting effect under stress.

### 3.1.2. Biochemical Analyses

Kaishi priming at dilutions of $10^{-9}$ and $10^{-12}$ resulted in increased chl *a* and chl *b* contents. There was no significant change in the carotenoid levels (Figure 2A). Heat stress decreased both chl *a* and chl *b* content in the non-primed plants and in the biostimulant-treated ones. However, priming with dilutions of $10^{-9}$ and $10^{-12}$ resulted in significantly higher chl *a* and chl *b* levels in the heat-stressed plants compared to the non-primed stressed plants. The chl *a*/*b* ratio remained stable in the groups of plants treated with $10^{-6}$, $10^{-9}$, and $10^{-12}$ biostimulant solutions, while in the non-treated plants subjected to heat stress, as well as in those that were treated with the $10^{-3}$ dilution of Kaishi, a reduction in the parameter was detected.

The amino acid proline is frequently used as a suitable stress marker for osmotic and combined drought and heat stress; therefore, we monitored the changes in its levels in the different treatment groups after heat weave exposure. Free L-Pro did not increase as a result of the applied stress in the plants without biostimulant pre-treatment (Figure 2B). However, the biostimulant promoted significant proline accumulation in the leaves of the heat-stressed plants in all test groups, except in the one that was sprayed with the $10^{-3}$ Kaishi dilution.

Malondialdehyde is a marker for oxidative damage to membranes. Its levels were significantly increased in the plants which experienced heat stress (Figure 2C). The MDA content in the heat-stressed individuals, which received biostimulant treatment, remained

close to the respective controls, and in the $10^{-9}$ test group, the measured levels were even lower than the ones in the non-stressed plants.

The analysis of some metabolites (Table 2) revealed that heat stress drastically diminished starch content and slightly decreased soluble sugars and flavonoids in biostimulant non-treated plants, whereas total phenolics did not change significantly under stress. Leaf starch content in the controls was positively influenced by Kaishi dilutions of $10^{-3}$ and $10^{-6}$. Soluble sugars were slightly diminished by Kaishi priming. The biostimulant stabilized leaf sugar and starch content under heat stress in all applied dilution series, which could be due to its indirect positive effect on photosynthetic processes.

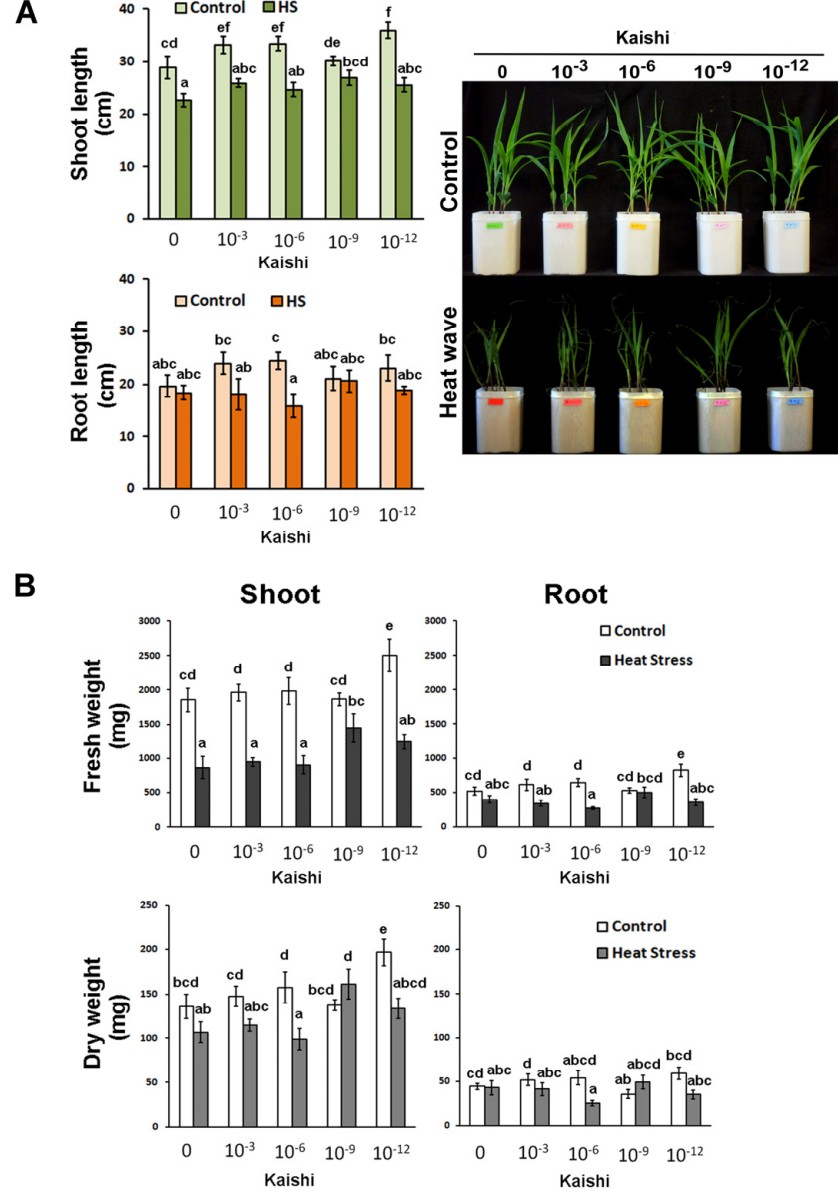

**Figure 1.** Effects of simulated heat wave on soil-grown maize plants treated with $10^{-3}$, $10^{-6}$, $10^{-9}$, and $10^{-12}$ biostimulant solution via a foliar spray. (**A**) Shoot length, root length, and visible status; (**B**) FW and DW of shoots and roots of differently treated plants. Values are means $\pm$ SD ($n \geq 15$). The different lowercase letters indicate statistically significant differences at $p < 0.05$ (multifactor ANOVA).

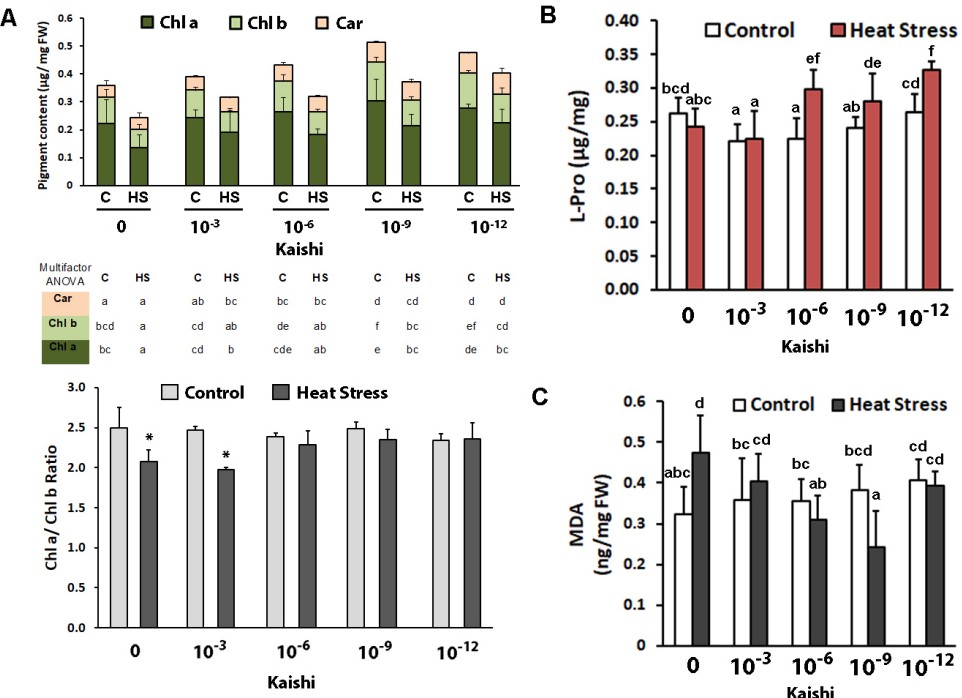

**Figure 2.** Photosynthetic pigments (**A**), free L-Pro content (**B**), and MDA (**C**) in soil-grown maize plants subjected to heat waves with or without biostimulant treatment (application via foliar spray of $10^{-3}$, $10^{-6}$, $10^{-9}$, and $10^{-12}$ Kaishi solution). Values are means of three biological repeats ($n = 3$) ± SD. Different lowercase letters (multifactor ANOVA) and asterisks (one-way ANOVA) indicate statistically significant differences at $p < 0.05$.

**Table 2.** Content of soluble sugars and starch (as glucose equivalents), total phenolics (as caffeic acid equivalents), and flavonoids (as rutin equivalents) in the leaves of Kaishi-pre-treated (K) and non-primed plants (K0) under control conditions (C) and subjected to heat stress (HS). Values are mean ± standard deviation from three replicates. Different lowercase letters indicate statistically significant differences at $p < 0.05$ (multifactor ANOVA).

| Priming | Treatment | Soluble Sugars (mg g$^{-1}$ FW) | Starch (mg g$^{-1}$ FW) | Total Phenolics (mg g$^{-1}$ FW) | Flavonoids (mg g$^{-1}$ FW) |
|---|---|---|---|---|---|
| K0 | C | 39.08 ± 2.32 e | 36.58 ± 3.44 f | 12.15 ± 0.26 ab | 4.47 ± 0.28 c |
|  | HS | 35.49 ± 2.19 cd | 10.68 ± 0.38 a | 12.9 ± 0.82 bc | 4.01 ± 0.12 a |
| K10$^{-3}$ | C | 31.46 ± 0.88 ab | 51.96 ± 4.59 h | 11.26 ± 0.44 a | 4.16 ± 0.15 ab |
|  | HS | 33.87 ± 1.76 bcd | 33.81 ± 1.88 def | 13.55 ± 0.21 c | 4.49 ± 0.29 c |
| K10$^{-6}$ | C | 35.48 ± 2.32 cd | 41.40 ± 2.45 g | 13.09 ± 0.15 bc | 4.29 ± 0.19 bc |
|  | HS | 35.87 ± 3.59 cde | 33.05 ± 1.69 cde | 13.55 ± 0.59 c | 4.14 ± 0.22 ab |
| K10$^{-9}$ | C | 32.97 ± 3.05 abc | 30.33 ± 1.57 bcd | 13.38 ± 0.40 c | 4.73 ± 0.18 d |
|  | HS | 33.92 ± 1.64 bcd | 34.23 ± 1.17 bc | 12.18 ± 0.22 a | 4.28 ± 0.24 bc |
| K10$^{-12}$ | C | 29.78 ± 2.39 a | 28.33 ± 1.18 ef | 13.08 ± 0.49 bc | 4.73 ± 0.11 d |
|  | HS | 37.15 ± 2.5 de | 30.87 ± 0.45 b | 14.83 ± 0.73 d | 4.10 ± 0.09 ab |

### 3.1.3. In-Gel Staining of Antioxidative Enzyme Activities

Secondary oxidative stress usually develops under prolonged or severe primary stresses, including heat waves. The increased ROS production is counteracted by different detoxifying enzymes, such as SOD, CAT, and various POX enzymes. Seven SOD isoforms, six POX, and one CAT isoform were detected via in-gel staining in the leaf samples (Figure 3). Under heat stress, an increase in the total SOD activity was observed, while the total POX activity remained unchanged. CAT activity diminished in the samples derived

from the heat-stressed individuals. In the Kaishi-treated plants, a slight increase in SOD and CAT activities was observed with the $10^{-6}$ dilution, while POX activity tended to be stimulated with the $10^{-9}$ dilution. The heat stress slightly induced further POX activity in the biostimulant-treated plants with the $10^{-6}$ dilution. The SOD activity marked its highest levels in the samples pre-treated with the $10^{-3}$ dilution. The applied stress reduced CAT activity with the $10^{-9}$ dilution.

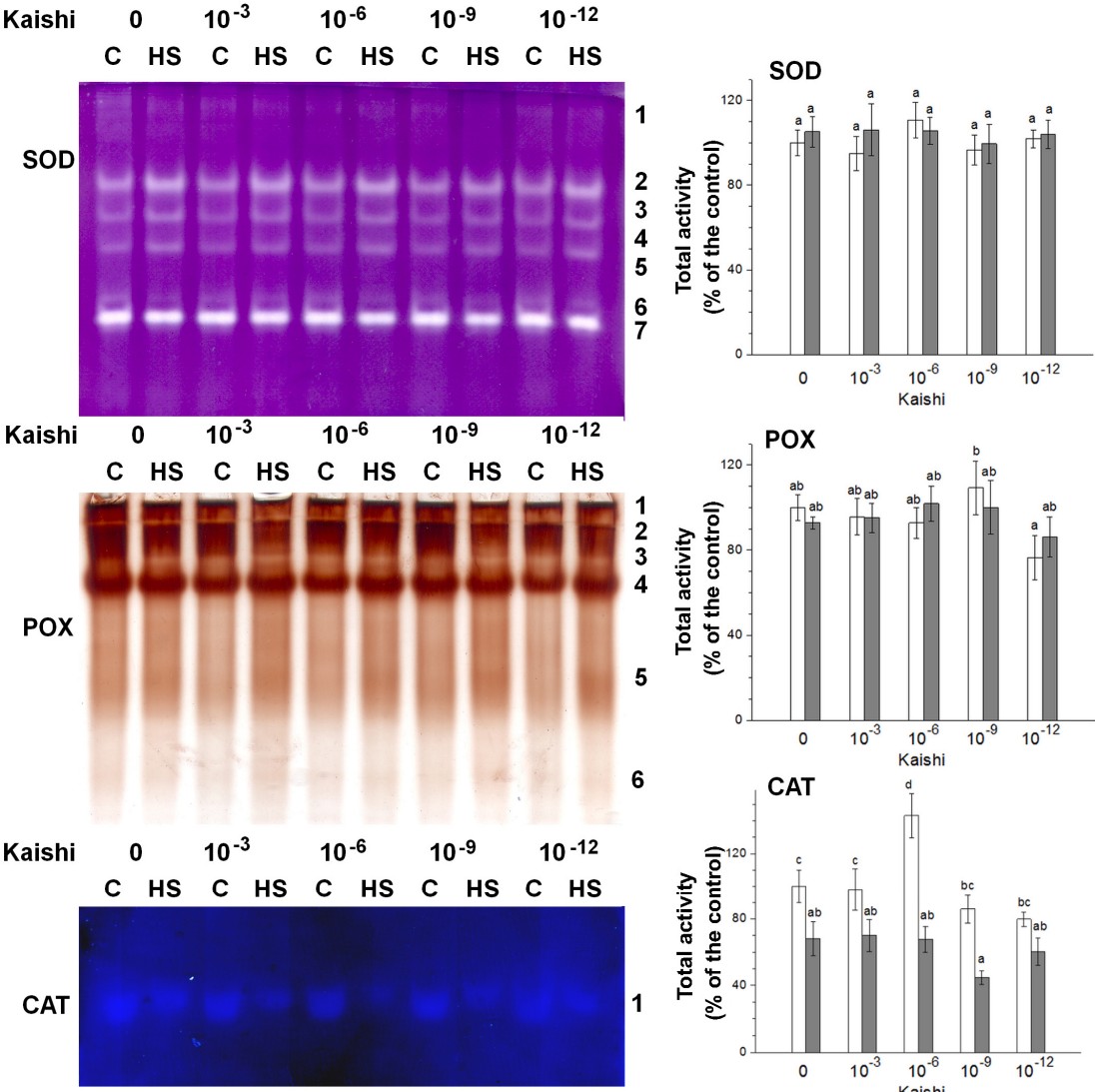

**Figure 3.** In-gel activity staining of SOD (40 µg of protein loaded per each lane), POX (20 µg of protein loaded per each lane), and CAT (60 µg of protein loaded per each lane) antioxidant enzymes visualized by representative gel images. The numbers along the right side of the gels indicate the positions of the distinct isoforms. The graphs represent mean values of three separate measurements of the staining intensity performed with ImageJ ($n = 3 \pm$ SE). The results are expressed as % of the control. White columns indicate the measured total enzyme activity in the non-stressed variants, and the grey columns depict the activity in the heat-stressed individuals from the same treatment group. Different letters above the columns indicate significant differences at $p < 0.05$ (multifactor ANOVA).

### 3.2. Growth-Stimulating Effects of the Biostimulant in Hydroponic Cultures

The effect of the same dilutions as the ones in the soil experiment ($10^{-3}$, $10^{-6}$, $10^{-9}$, and $10^{-12}$) on seed germination (72 h in darkness), organ growth, and biomass accumulation during the first seven days was assessed (Figure 4). The biostimulant, applied in the range of the recommended concentrations for agricultural practice ($10^{-3}$), suppressed the seed

germination in darkness within the first 72 h (Figure 4A). The growth-stimulating effect of Kaishi added to the media in lower amounts was observed later, when the plants were transferred into a growth chamber with a 16/8 h photoperiod (Figure 4A).

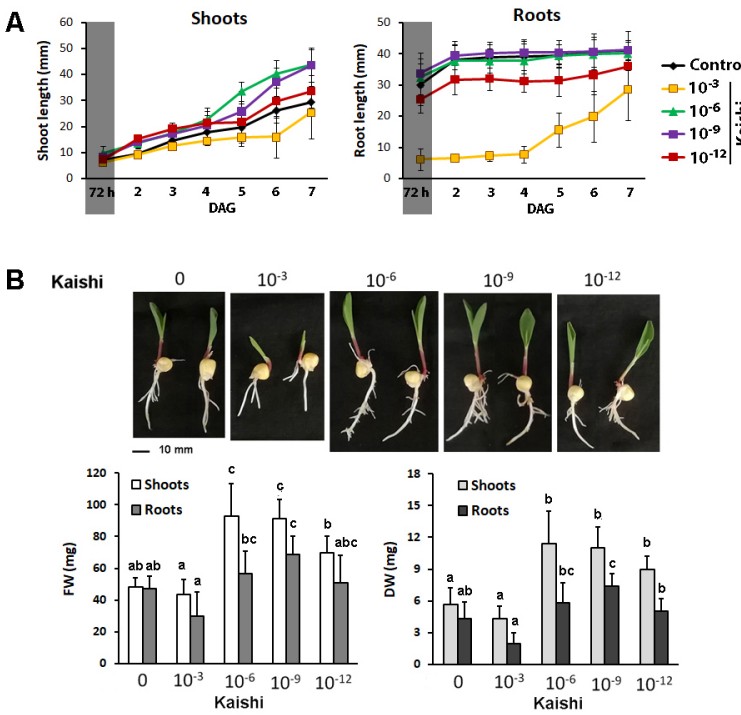

**Figure 4.** Effects of different amounts of the biostimulant on plant germination and growth in hydroponic cultures. (**A**) Germination for 72 h in darkness (reflected by a shaded sector on the graph) and organ growth on Kaishi containing media; (**B**) phenotype, fresh weight (FW), and dry weight (DW) of shoots and roots of 7-day-old plants on Kaishi containing media. The different lowercase letters designate statistically significant differences at $p < 0.05$, and the error bars represent SD ($n \geq 20$, multifactor ANOVA).

The $10^{-3}$ biostimulant dilution delayed the root elongation compared to the other test groups (Figure 4A) and did not provoke significant biomass changes (Figure 4B). The lower amounts of the biostimulant ($10^{-6}$, $10^{-9}$, and $10^{-12}$) supplemented to the nutrient solution promoted shoot and root growth (Figure 4A) and resulted in increased fresh and dry weight of the aboveground part of the treated plants (Figure 4B). The root-biomass-stimulating effect was also detected, and this effect was most pronounced in the $10^{-9}$ treatment group.

*3.3. Pre-Treatment with Kaishi Mitigates the Negative Effect of Heat Shock in Hydroponically Grown Maize Plants*

To determine possible molecular targets of the product action, we employed a hydroponic experiment with strictly controlled growth parameters using the highest Kaishi dilution ($10^{-12}$). The imposed heat shock resulted in statistically significant changes in the monitored growth parameters, evident by the shorter shoots and roots, and the decreased dry weight of the organs (Figure 5). The application of the biostimulant at $10^{-12}$ dilution had a stress-mitigating effect manifested by the relatively unaffected organ growth, and the higher FW and DW of the pre-treated and subsequently heat-stressed plants, as compared to the stressed individuals, which did not receive biostimulant priming.

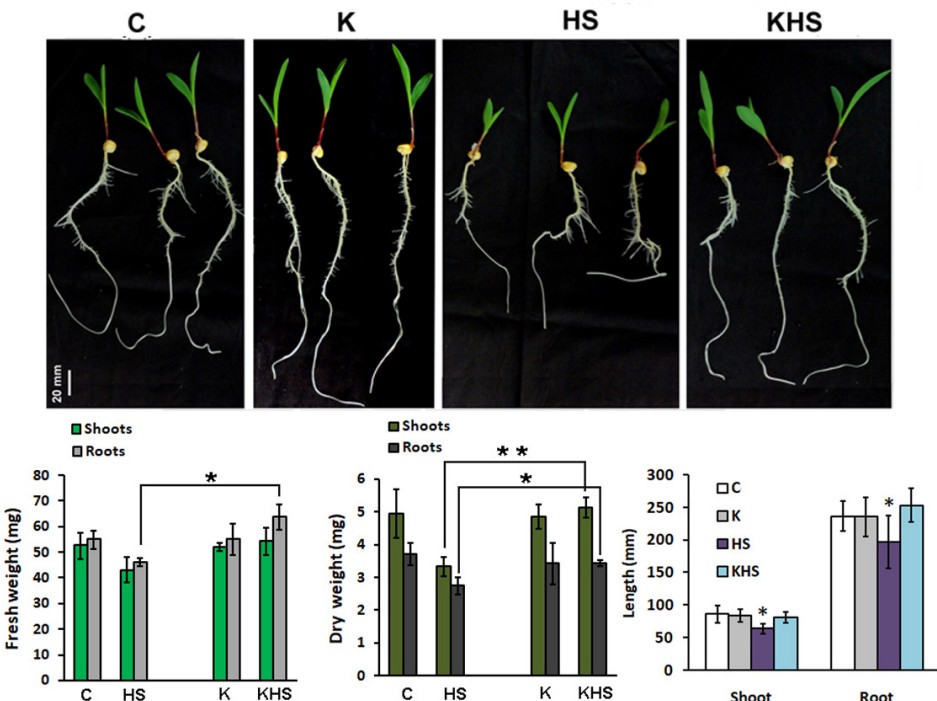

**Figure 5.** Phenotype and growth parameters (organ length, FW, and DW) of 11-day-old hydroponically grown maize plants, 48 h after the applied heat shock (1 h at 45 °C). C, control; HS, heat shock; K, treated with Kaishi (dilution $10^{-12}$) and grown under normal conditions; KHS, treated with Kaishi (dilution $10^{-12}$) and subjected to heat shock 24 h later. Asterisks designate statistically significant differences at $p < 0.05$ (*) and $p < 0.01$ (**). The error bars represent SD ($n \geq 15$, one-way ANOVA).

*3.4. Protease Activity in Plants Treated with the Biostimulant and Subjected to Heat Shock*

The strong induction of HSPs and DHNs by high temperature and drought stress is well established, but considerably less information can be found about protease's response to heat shock. We analysed some protease activities 48 h after the heat shock application (HS) via in-gel staining in two preliminarily established pH optima—acidic (pH 5.0), which mainly reveals cysteine-type proteases, and neutral pH (pH 7.5), which mainly visualizes the activity of serine-type proteases (Figure 6). Activity staining showed distinct organ-specific profiles. In the root samples, up to 10 activity bands were revealed at pH 5.0 and 3 bands at pH 7.5. In the leaf samples, four activity bands at pH 5.0 and two bands at pH 7.5 were identified. The applied heat shock resulted in a slight increase in the total proteolytic activity at both pH conditions. Kaishi pre-treatment had also a protease-activity-stimulating effect. However, the heat stress applied on the biostimulant-primed plants provoked some protease activity reductions, particularly under acidic conditions.

*3.5. Transcript Profiling of HSP-, DHN-, and Protease-Coding Genes in Plants Treated with the Biostimulant and Subjected to Heat Shock*

The expression of certain specific genes from the groups of HSPs, DHNs, and proteases in samples collected 24 h after the heat shock were monitored (Figure 7). These groups of proteins are known to be actively involved in the protection of macromolecules from the deleterious effects of heat stress. The transcript profiling was performed using published mRNA complete sequences in the NCBI database (accessed in July 2019).

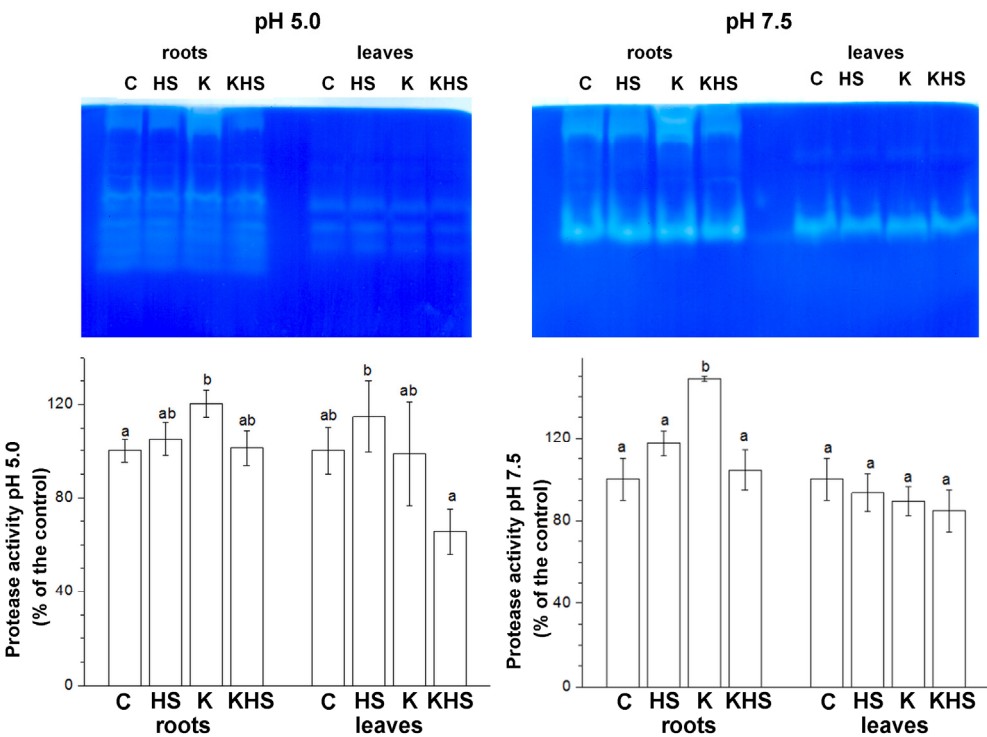

**Figure 6.** In-gel staining of acidic (pH 5.0) and neutral (pH 7.5) protease activity. Protein loading on each lane equals 25 µg of total soluble protein for the root samples and 50 µg of total soluble protein for the leaf samples. The graphs represent mean values of three separate measurements of the staining intensity performed with ImageJ ($n = 3 \pm$ SE). The results are expressed as % of the control. Different letters above the columns indicate significant differences at $p < 0.05$ (multifactor ANOVA).

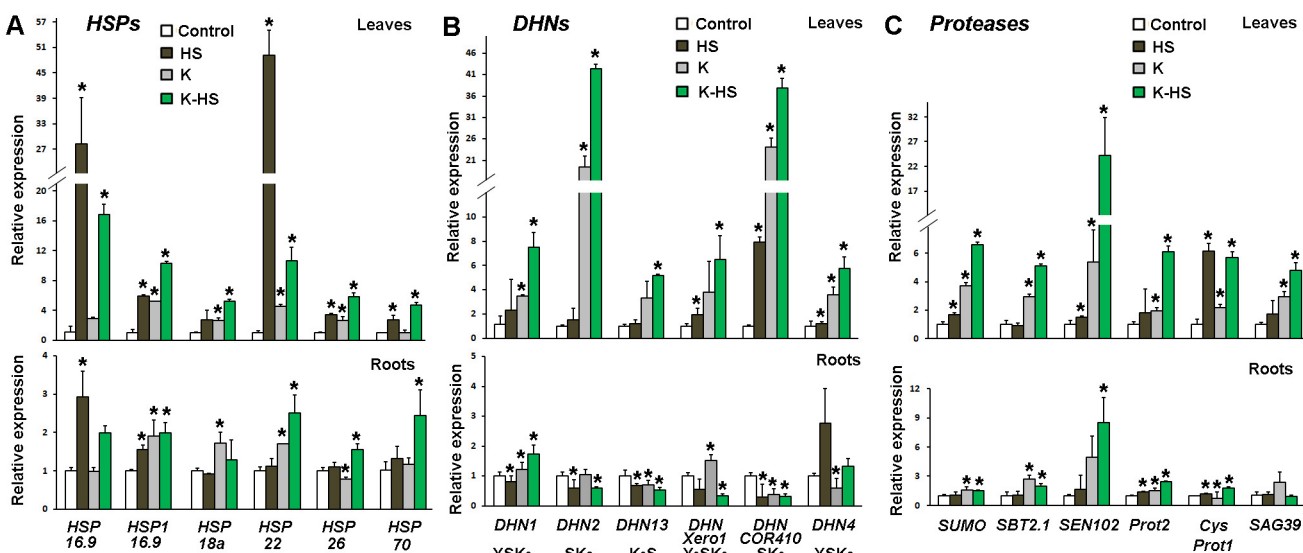

**Figure 7.** Transcript profiling of genes coding for heat shock proteins (HSPs) (**A**), dehydrins (DHNs) (**B**), and proteases (**C**) in leaves and roots of control and heat-stressed plants (HS) which were not treated with biostimulant; plants which received Kaishi ($10^{-12}$) and were grown under normal conditions (K) or subjected to heat shock 24 h later (K-HS). Values are means of three biological repeats ($n = 3$) $\pm$ SD, and the asterisks designate statistical significance compared to the control sample at $p < 0.05$, (one-way ANOVA).

Six *HSPs* were analysed for transcript abundance changes in the leaves and roots of control and Kaishi-pre-treated maize seedlings subjected to HS; among them, five small

*sHSPs* (*HSP 16.9*, *HSP1 16.9*, *HSP 18a*, *HSP 22*, and *HSP 26*) and one *HSP 70* were included. The gene expression levels increased in the heat-stressed leaves, with *HSP 16.9* and *HSP 22* showing the greatest changes in transcript abundance. *HSP 16.9* and, to a lesser extent, *HSP1 16.9* transcripts predominantly accumulated in the roots of the heat-stressed primed individuals. Kaishi treatment alone increased the basal level of all tested *sHSP* genes both in the leaves and in the roots, which could be considered a priming effect. Kaishi pre-treatment alone strongly stimulated the transcription of *HSP1 16.9*, *HSP 18a*, and *HSP 22* in the plants grown under normal conditions. Heat stress further induced the activation of the transcription of some of the genes in the biostimulant pre-treated plants (*HSP 16.9*, *HSP 18a*, and *HSP 26* in the leaves, and *HSP 22*, *HSP 26*, and *HSP 70* in the roots).

The stress treatment resulted in the increased expression of *DHN* genes with *DHN-COR410* (coding for an SK2-type dehydrin) and *DHN4* (YSK3 type) reaching particularly high levels, respectively, in the leaf and in the root samples. The pre-treatment with the biostimulant induced the leaf expression of most of the tested *DHN* genes, with *DHN2* (SK3 type) and *DHNCOR410* (SK2 type) showing the most prominent upregulation. The stress imposed on the biostimulant pre-treated plants further increased the *DHN* transcript levels in the leaves. In the roots of the pre-treated plants under heat stress, increased expression was only observed for *DHN1* (YSK2 type) and *DHN4* (YSK3 type) transcripts.

The analyses of the protease-coding gene expression showed that in the non-primed plants, the imposed high temperature treatment strongly induced only *CysProt1* transcript accumulation in the leaves. However, Kaishi priming seemed to elevate the basal transcript level of all studied protease genes in the leaves, and it also induced, to some extent, the accumulation of *SBT2.1*, *SEN102*, and *SAG39* transcripts in the roots. The heat stress imposed on the biostimulant pre-treated individuals additionally increased the relative expression of all the protease-coding genes in the leaves. The combined treatment similarly affected the gene expression in the roots, with particularly higher levels documented for *SEN102* transcripts.

## 4. Discussion

Biostimulants operate through still not well characterized mechanisms that differ from the effects of conventional fertilizers. Understanding the processes involved in their protective action will deliver the necessary knowledge for the development of precise application protocols targeting different crops to mitigate the effects of specific stress factors. Over the last few years, a number of studies have testified that along with their growth-accelerating properties, biostimulants have shown protective potential against abiotic stress which is dependent on the crop type and the application approach [12,20,49–53]. Here, we present data demonstrating the stress-mitigating effects of a biostimulant applied at a nanoscale level which could be utilized in sustainable crop management strategies to mitigate the effect of adverse environments. The accumulated research-based proof for the efficient use of biostimulants in agricultural practice corresponds to the goals set by the European Green Deal and the Farm to Fork strategy of the current EU policy for sustainable agricultural ecosystems (https://ec.europa.eu/info/strategy/priorities-2019-2024/european-green-deal_en; https://ec.europa.eu/food/horizontal-topics/farm-fork-strategy_en, accessed on 12 April 2022).

A recently published study employed a higher concentration of Kaishi for seed pre-treatment and subsequent testing for stress-mitigating effects [27]. Starting with a dilution recommended by the supplier and comparing two modes of application (foliar spray and uptake through roots from the nutrient medium) we aimed to determine the useful concentration range for general stimulating and stress-protective effects of the biostimulant. Protein-hydrolysate-based biostimulants such as Kaishi can be absorbed directly through leaves and evidently more effectively through the root system [15]. The obtained results using a range of the biostimulant dilutions attest that the growth stimulation seems to be associated with improved mineral nutrition, photosynthetic performance, and the better preservation of cell structures and proteins. These findings are in line with previously pub-

lished observations which also discussed the interference of the biostimulant application with the hormonal balance and the upregulation of specific transcripts [15,24–26].

The present study also demonstrates that biostimulant priming mitigates the negative effects of heat stress. Its application maintained plant growth, stabilized pigment content and the chl $a/b$ ratio, and restored the starch content to the level of control plants. The stabilizing effect of Kaishi priming was observed both after foliar and after root application. The slight but significant increase in leaf pigment content after Kaishi pre-treatment for all applied dilutions, with the best results for $10^{-9}$, as well as more starch accumulation in leaves (as a storage product from photosynthesis) after spraying with the $10^{-3}$ and $10^{-6}$ dilutions, provided some evidence for improved photosynthetic performance. The observed positive effects of the higher concentrations of the biostimulant are to be expected as the product has been obtained through enzymatic hydrolysis, which delivers a complex of eighteen readily available amino acids that could be directly assimilated by plants. This particular characteristic of the product enables an increase in energy to support or resume growth in the event of stress.

The detected drop in the chl $a/b$ ratio under heat stress indicates a greater reduction in chl $a$, which attests to obstructed interaction between the Photosystem II cores and the light-harvesting chlorophyll–protein complex, where the major part of chl $b$ is present. Therefore, the stabilized chl $a/b$ levels suggests that the applied biostimulant had an indirect protective effect on the photosynthetic apparatus.

The mobilization of ROS-scavenging enzymes such as SOD, CAT, and POX is a common heat stress counteracting mechanism [54]. The heat stress effects on the non-primed soil-grown maize plants caused significant increases in lipid peroxidation and SOD activity and a diminution in CAT activity. These changes indicated the occurrence of oxidative damage to membranes and the development of oxidative stress, confirming previously published reports [55]. SOD and POX enzymes remained stable in Kaishi-pre-treated plants upon heat stress. The relatively lower MDA levels in the biostimulant-primed plants subjected to high temperature stress observed in the present study suggests that the pre-treatment with Kaishi has a protective effect, as it reduces lipid peroxidation and the oxidative damage of the cellular membranes.

The priming with Kaishi was beneficial in counteracting the heat stress consequences, as it provoked a significant increase in proline content. Besides being a well-known osmo-protectant, free proline also possesses ROS scavenging, protein-protecting, and signalling properties [56,57].

The capability of the biostimulant Kaishi to exert efficient priming at high dilutions prompted us to analyse its effects on the transcript accumulation of several genes coding HSP, dehydrin, and proteases. Hydroponic cultures were preferred for these analyses due to the direct and efficient uptake of the biostimulant by the root system.

The important role of the small HSPs and HSP 70 in protein repair motivated the gene expression analyses of published full genetic sequences of the corresponding genes in the NCBI database. HSPs confer protein and membrane stability, leading to improved photosynthesis, better assimilate partitioning, and the more efficient use of water and nutrients [33]. Small HSPs in plants are very abundant, diverse, and are distributed in every cell compartment. These ATP-independent molecular chaperones form dynamic oligomeric complexes that bind denatured proteins and prevent irreversible protein aggregation. Next, with the help of sHSPs, they can be refolded in cooperation with ATP-dependent chaperones such as the HSP 70 complex [58], or targeted for degradation by proteases. There are a number of sHSP subfamilies (up to 10 in monocots). Some of them are constitutively expressed and others are highly upregulated in response to heat and other stresses [59]. In our study, we analysed the transcript abundance of five different *sHSPs* as well as of an *HSP 70*-coding transcript in control and Kaishi-pre-treated maize seedlings under normal conditions and after HS. Kaishi treatment alone increased the basal level of all of them except *HSP 70*, and the applied heat stress resulted in the additional activation of their transcription, especially for *HSP 16.9* and *HSP 22* in leaves and for *HSP 16.9* and *HSP1*

*16.9* in roots (Figure 7A). The small HSP 16.9 has cytoplasmic/nuclear localization, while HSP 22 is mainly found in mitochondria and HSP 26—in chloroplasts [59–61]. The different transcript activation could reflect the diverse protective functions of the various sHSPs at different subcellular locations. For example, interaction of HSP 26 with specific photosynthetic proteins has been described [61]. The similar dramatic induction of maize mitochondrial HSP 22 under HS has been previously reported by [60]. Rashed et al. (2021) [34] linked the strong upregulation of several *HSP* genes in maize, including *ZmHSP 16.9*, *ZmHSP 22*, and *ZmHSP 70*, to greater heat tolerance of the tested maize lines. Our results demonstrate the positive effect of Kaishi biostimulant priming, which provokes higher *sHSPs* basal levels and their stronger subsequent induction upon heat stress.

DHNs are another type of protective proteins that have a wide range of functions and can be induced by a variety of conditions, including drought, salt, and abnormal temperatures [31,62]. They are highly hydrophilic and thermostable intrinsically disordered proteins, with largely varying molecular weight (9–200 kDa). All of them contain a lysine-rich K segment with repeated glycine and polar amino acids forming amphipathic helices, which could interact with lipids and hydrophobic sites of partially denatured proteins, stabilize them, and protect them from denaturation [63]. DHNs are able to bind water and metal ions (via their His-rich domain), nucleotides (through the Y-segment), and also, they could be phosphorylated (particularly the S-segment). They could even operate as antioxidants capable of scavenging hydroxyl radicals [39,62]. The interaction of dehydrins with different ligands such as metals, biomembranes, and proteins protects macromolecules from the deleterious effect of heat stress, which often directly causes protein denaturation and increases membrane fluidity. The association between DHN accumulation and plant tolerance to heat stress has been previously evidenced [31]. A set of 19 putative maize DHN genes has been identified in silico, and the characterization of their promoter regions showed that some of them contained heat shock elements [63]. Similarly to HSPs, DHNs are also distributed in various parts of the cell—the cytoplasm, nucleus, plasma membrane, tonoplast, plastid, mitochondrion, and endoplasmic reticulum. In a published study on maize dehydrins [64], the authors identified DHN family members with cytoplasmic (*ZmDHN1* and *ZmDHN4*) and nuclear localization (*ZmDHN2* and *ZmDHN13*). The induction of *ZmDHN13* by low temperature, ABA, and the excessive application of copper, osmotic, and oxidative stress has been reported [64]. In our study, we analysed the effect of the biostimulant priming and HS on the expression of a set of six *ZmDHN* genes coding SK and YSK type dehydrins. The *DHN* response to heat stress was not as strong as that of *sHSPs*, and there were variances depending on the DHN type and the plant organ. Heat stress upregulated an SK-type *DHN* gene (*DHN COR410*) in the leaves and induced the expression of an YSK-type *DHN* gene in the roots (although the result appeared to not be statistically significant) (Figure 7B). The biostimulant priming increased the abundance of all examined DHN transcripts in the leaf samples but induced the levels of only one in the roots. The heat shock applied to Kaishi-treated plants additionally stimulated the accumulation of all *DHN* transcripts in the leaves, whereas only *DHN1* and *DHN4* (both of them coding YSK-type dehydrins) were upregulated in the root samples after the combined treatment. These results suggest that the biostimulant priming increases the basal levels of *DHN* transcripts, which could be beneficial for plants experiencing different environmental challenges.

The monitoring of protease activity in plants under stress reports on the degradation of proteins that could not be repaired. The degradation of unnecessary proteins in conditions of depressed photosynthesis and carbon starvation under stress could be also a source of substrates reinvested to fuel the plant metabolism. [33]. Plants exposed to temperature and osmotic stress have been found to accumulate high levels of mRNAs that encode proteases, particularly cysteine and serine types [65]. In research on bentgrass genotypes with diverse heat tolerance, the authors reported that the heat-sensitive line had a higher level of protease activity than the tolerant one [66]. The small increase in the acidic protease activity in leaves and roots as a result of biostimulant priming—at both examined pH optima—could

be linked to possible regulatory action exerted by protein-hydrolysate-based products. Our experiments demonstrated that the acidic protease activity was maintained in roots while it diminished in leaves, whereas neutral protease activity diminished in roots and was not changed in leaves. This stability or downregulation of proteolytic activities after the applied heat stress could be due to the lower number of damaged proteins in the primed plants. To evaluate the priming effects on the protease transcript accumulation, we analysed the expression of six protease genes in response to the biostimulant pre-treatment and the imposed HS. The gene coding for SUMO protease enzyme was included in the analyses as it has been substantially activated by high temperatures and drought in maize [67,68]. Two of the other protease genes under investigation were serine-type proteases (*Prot2*—coding for a B-like oligopeptidase, and *SBT2.1*—coding for a subtilisin type protease). We also analysed the expression of three cysteine protease-coding genes (*CysProt1*, *SEN102*, and *SAG39*) [7]. We found a positive relationship between the biostimulant priming and the activation of the tested protease genes, particularly in the leaves of the heat-stressed individuals. Among them, *SEN102* transcripts, coding for a cysteine-type protease, marked the highest induction, a trend which was also observed in the root samples. Both the transcript profiling and the activity staining patterns point to a complex regulation of proteases under heat stress in an organ-specific context. It should be noted that the documented trends in the transcript level changes and the total protease activity are not expected to always match due to the high level of diversity of the protease enzyme families as well as the complex regulation and precise control of the various protease activities operating in the plant cell.

## 5. Conclusions

The obtained results provide insights into the mechanism of action of the biostimulant Kaishi and its application as a priming agent, able to mitigate heat stress in maize during early vegetative development. In addition to the expected beneficial effects on growth, photosynthesis stabilization and better oxidative stress management, the presented study demonstrates the enhanced organ-specific accumulation of transcript coding for protective proteins such as HSPs, dehydrins, and proteases. Overall, the transcript activation of these genes is expected to be beneficial for mitigating various types of abiotic stresses. Additional research on the ameliorating activity towards other adverse environmental factors in field conditions is necessary to optimize the application protocols. The use of very low amounts of the biostimulant that still allows for better survival and recovery rates of the stress-affected crop to be achieved is in line with sustainable agricultural approaches and could be of particular importance for the reduction in the final product costs.

**Author Contributions:** Conceptualization, I.I.V. and V.V.; methodology, I.I.V. and L.S.-S.; validation and formal analysis, I.I.V. and L.S.-S.; investigation, I.I.V., L.S.-S., T.K., A.K. and B.Y.-M.; writing—original draft preparation, I.I.V. and L.S.-S.; writing—review and editing, L.S.-S. and V.V.; visualization, I.I.V. and L.S.-S.; funding acquisition, I.I.V., L.S.-S., and V.V. All authors have read and agreed to the published version of the manuscript.

**Funding:** This research was funded by the Bulgarian Ministry of Education and Science under the National Research Programme "Healthy Foods for a Strong Bio-Economy and Quality of Life" approved by DCM # 577/17.08.2018.

**Conflicts of Interest:** The authors declare no conflict of interest.

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
