# Peer review of "Heat-Stress-Mitigating Effects of a Protein-Hydrolysate-Based Biostimulant Are Linked to Changes in Protease, DHN, and HSP Gene Expression in Maize"

_agronomy, doi:10.3390/agronomy12051127_

Round 1

Reviewer 1 Report

A well-written and interesting story. I had no idea a dilute spray of basically amino acids could have such protective value. It is a relevant finding to environmentally conscious agriculture.

I have made minor suggested edits to the text marked in red in the attached PDF. 

My only criticism (probably due to my own ignorance) is with the method by which you measured activities of the antioxidant enzymes. Can one be certain that this method accurately measures these activities? Differences are not obvious in Fig. 3 nor is significance or error indicated in the graphs which you expressed from the gel images "using Image J software to estimate total enzyme activity." Perhaps some statement to qualify confidence in these measures is needed. Does the cited methodology mention reliability of such measures? I have a similar issue with your protease activities shown in Fig. 6. In Fig. 6 the "leaves" label is missing on the first graph (pH 5.0).

Neither of these criticisms kill my recommendation to publish your paper. Generally I believe the differences you report, but anything you can say to convince your readers that the enzyme activity measures are reliable and real will improve your message.

Very nice work!  I enjoyed reading your paper.

Author Response

A well-written and interesting story. I had no idea a dilute spray of basically amino acids could have such protective value. It is a relevant finding to environmentally conscious agriculture.

I have made minor suggested edits to the text marked in red in the attached PDF. 

Answer: We are very grateful for the positive feedback! We tried to address the critical comments on the in-gel activity staining results in the paragraphs below.

Unfortunately, the PDF with the suggested edits is not visible in the system so we were not able to follow them but we did try to improve the text (the changes in the revised version of the manuscript are marked in red through the Track Changes tool).

A small clarification on the biostimulant composition is due as we have missed correcting a typo in the previous version of the manuscript (page 2) - 92% should be 12% free amino acids. Besides the free L-amino acids, the product contains 2% organic nitrogen and the rest are miscellaneous derivatives (not disclosed by the manufacturer) originating from the enzymatic hydrolysis of standardized substrate of plant origin. Information on the composition of Kaishi is included in the revised version of the manuscript as suggested by one of the other reviewers.

My only criticism (probably due to my own ignorance) is with the method by which you measured activities of the antioxidant enzymes. Can one be certain that this method accurately measures these activities?

Answer: The in-gel staining gives a good comparative activity pattern of the different variants, rather than an estimation of the absolute enzyme activity. Sometimes it is indeed the preferred approach to assess the enzyme activities, especially in the cases in which the standard biochemical assays report unchanged total activity failing to disclose the diverse effects on the individual isoenzymes. The representative gel images in the figures (Figs. 3 and 6) show the activities of the distinct isoforms whereas the graphs were included to illustrate the total activity. The cited methodology describes well the reliability and the validation of these measurements.

Generally, the areas of the bands are proportional to the enzyme activities, and the ImageJ program has a special section for Gel analyses which is very suitable for estimation of the areas of individual bands after properly subtracting the background. The crucial part here is the equal loading of the lanes with one and the same amount of total soluble protein to ensure that the observed differences among the tested variants are due to the imposed treatments.

We are routinely using in-gel staining methods whenever it is possible, as they have the advantage of providing straightforward information on both the total activity and the isoenzyme patterns.

The principle of the activity staining approach is similar to the conventional assays for spectrophotometric determination of enzyme activities. The advantageous feature of this technique is that it successfully combines electrophoretic separation of the targeted enzymes and the measurement of their activity by using specific substrates.

(Hre are some details: In the in-gel staining protocol for SOD activity, we apply nitroblue tetrazolium which is also applied in the conventional biochemical SOD assay. For CAT activity measurements the gels are initially soaked in a solution containing hydrogen peroxide. After a certain period of incubation, the gels are subjected to “negative” staining which reveals only the areas on the gel where hydrogen peroxide has been inactivated by CAT. For the POX activity staining, we use benzidine as a substrate which gives an insoluble product as a result of the enzymatic reaction. This allows the product to be visualized on the exact sectors of the gel where the respective POX isoforms migrate during the native electrophoresis separation. For the estimation of the proteolytic activities, gels are co-polymerized with an appropriate protease substrate (in our case it is gelatin). After the electrophoretic separation, the proteases need to be reactivated by chasing away SDS with Triton X-100. After this, the gels are incubated at appropriate pH and stained with Coomassie to reveal the sectors on the gel where the substrate has been utilized by respective protease isoenzymes.)

Differences are not obvious in Fig. 3 nor is significance or error indicated in the graphs which you expressed from the gel images "using Image J software to estimate total enzyme activity." Perhaps some statement to qualify confidence in these measures is needed. Does the cited methodology mention reliability of such measures? I have a similar issue with your protease activities shown in Fig. 6.
Answer: To address the expressed concern we revised the graphs in Figs. 3 and 6, by presenting the results as mean relative values (percentage of the respective controls) of three separate measurements with the calculated standard error. The statistical significance of the measurements is also depicted on the revised bar charts. It was determined by applying multifactor ANOVA.

In Fig. 6 the "leaves" label is missing on the first graph (pH 5.0).

Answer: The missing label has been added to the figure.

Reviewer 2 Report

This manuscript investigated how Kaishi stimulant compounds affect maize response to heat stress. The authors indicated that the application of Kaishi may improve plants growth under high temperature stress. The title and subject of the manuscript are very interesting from a practical point of view, suitable and adequate. The study was well planned and performed and it collects a series of measures on the different biochemical parameters.
The scientific content contributes to the space in which it develops. Use of different species of new stimulant compounds is actual and modern topic in experimental research and scientific journals, however, application in real agriculture is still not very developed from the practical reasons. The introduction provides a good understanding of the subject and its importance, with a significant quantity of information. Theoretical and practical reasons for the study are very reasonable.
Very interesting results could be useful to verify in different environmental conditions to understand better plant adaptations to the changing environment and mitigating potential of the application of stimulant compounds.
-Several places in the materials and methods are not clear. The authors need to provide detailed information about the chemicals, such as their grade and purity.

Overall, the study is of good quality and the results are innovative. Thus the theoretical and practical reasons for this study are very reasonable. The manuscript is interesting, associated with actual plant science trends. This study presents the relevant matter in more depth than some of the other related publications. It extends previous findings by the different authors. The paper brings many new aspects, however, I would like to invite authors to discuss more eco-physiological aspects and molecular mechanisms using/reading new actual references.
The work was provided with a sufficient level of scientific novelty. This seems to be a well-conducted analysis of the literature and to have a clear, informative character.
The structure of the paper is logical and the results are well interpreted. The introduction and discussion are well organized. Conclusions are presented in an appropriate fashion and are supported by the data. I think the overall concept is interesting and potentially important. I should have expected a more critical discussion. Please check the comments and suggestions in the attached main text. I recommend to accept the paper for publication with minor revision after correction.

Author Response

We would like to thank the reviewer for the positive evaluation and for the made suggestions.

  1. “Please add the composition of Kaishi compound, it would be very important to bring some details to give more information for researcher and who want to use this material in future projects.”

Answer: According to the producer’s information the preparation Kaishi has the following Composition: Free amino acids -  12 % w/w/ (standard aminogram: L-glutamic acid, L-aspartic acid, L-alanine, L-arginine, L-cystine, L-phenylalanine, glycine, L-histidine, Lisoleucine, L-leucine, L-lysine, L-methionine, L-proline, L-serine, L-tyrosine, L-threonine, L-tryptophan, L-valine. No amino acid exceeds 20% of the total); Total Nitrogen (N) 2.0%w/w; Organic nitrogen 2.0%w/w. This information is included in the revised visions of the manuscript (Material and Methods, page 4) as suggested by the reviewer.

  1. “Regarding the figures, the treatment in high concentration could have any significant difference. Why? The authors should explain more with scientific reason in discussion part.”

Answer: To address the reviewer’s remark we include the following text in the Discussion section (page 16): “The observed positive effects of the higher concentrations of the biostimulant are to be expected as the product has been obtained through enzymatic hydrolysis which delivers a complex of eighteen readily available amino acids that could be directly assimilated by the plants. This particular characteristic of the product enables an increase in energy to support or resume growth in the event of stress.

Reviewer 3 Report

agronomy-1705824-peer-review-v1

The manuscript is a very important topic and informative, it has new knowledge.

A few points can be corrected by the authors:

The section on Material and Methods

2.1. Soil experiment ….this part is NOT written well, the authors should write down the concentrations of the biostimulant, usually we used the concentration-related percentage.

Also, Hydroponic experiments are NOT clear….the authors should explain how this experiment

2.3. Heat shock of hydroponically-grown plants also this part is NOT clear

Author Response

We would like to thank the reviewer for the positive evaluation and for the critical reading of the text regarding the description of the experimental setups.

The manuscript is a very important topic and informative, it has new knowledge.

A few points can be corrected by the authors:

The section on Material and Methods

2.1. Soil experiment ….this part is NOT written well, the authors should write down the concentrations of the biostimulant, usually we used the concentration-related percentage.

Also, Hydroponic experiments are NOT clear….the authors should explain how this experiment

2.3. Heat shock of hydroponically-grown plants also this part is NOT clear

Answer: We carefully revised the text describing the experimental setups in the M&M section and tried to improve its comprehensibility. The equivalent of the starting dilution in % has been also included in the text (on page 4): “As a starting concentration, we used an amount (1 ml/L or 10-3 dilution which corresponds to 0.001% solution) that approximates the range recommended by the producer (1-3 l/ha).”